# Pitfalls in Single Clone CRISPR-Cas9 Mutagenesis to Fine-Map Regulatory Intervals

**DOI:** 10.3390/genes11050504

**Published:** 2020-05-04

**Authors:** Ruoyu Tian, Yidan Pan, Thomas H. A. Etheridge, Harshavardhan Deshmukh, Dalia Gulick, Greg Gibson, Gang Bao, Ciaran M Lee

**Affiliations:** 1Center for Integrative Genomics, Georgia Institute of Technology, Atlanta, GA 30332, USA; rtian@gatech.edu (R.T.); dalia.arafat@biology.gatech.edu (D.G.); 2Systems, Synthetic, and Physical Biology, Rice University, Houston, TX 77005, USA; yp11@rice.edu; 3Department of Bioengineering, Rice University, Houston, TX 77005, USA; hd16@rice.edu (H.D.); thomasetheridge@alumni.rice.edu (T.H.A.E.); 4APC Microbiome Ireland, University College Cork, Cork T12 YN60, Ireland

**Keywords:** eQTL, CRISPR-Cas9, single-cell clone, fine-mapping, power

## Abstract

The majority of genetic variants affecting complex traits map to regulatory regions of genes, and typically lie in credible intervals of 100 or more SNPs. Fine mapping of the causal variant(s) at a locus depends on assays that are able to discriminate the effects of polymorphisms or mutations on gene expression. Here, we evaluated a moderate-throughput CRISPR-Cas9 mutagenesis approach, based on replicated measurement of transcript abundance in single-cell clones, by deleting candidate regulatory SNPs, affecting four genes known to be affected by large-effect expression Quantitative Trait Loci (eQTL) in leukocytes, and using Fluidigm qRT-PCR to monitor gene expression in HL60 pro-myeloid human cells. We concluded that there were multiple constraints that rendered the approach generally infeasible for fine mapping. These included the non-targetability of many regulatory SNPs, clonal variability of single-cell derivatives, and expense. Power calculations based on the measured variance attributable to major sources of experimental error indicated that typical eQTL explaining 10% of the variation in expression of a gene would usually require at least eight biological replicates of each clone. Scanning across credible intervals with this approach is not recommended.

## 1. Introduction

Genome-wide association studies (GWAS) over the past decade have been highly successful in identifying tens of thousands of loci influencing disease risk [1,2,3], but the fine mapping of causal variants has failed to keep pace. Exhaustive studies of Crohn’s disease and type 2 diabetes associations, for example, indicate that the average credible interval size for hundreds of loci remains over 100 SNPs, and fewer than 15% of the loci have been reduced to a single high-confidence causal polymorphism [4,5]. This gap in knowledge impedes both the understanding of the biological functions of risk loci and the progress in clinical genetic risk assessment. There are three main challenges to fine mapping. First, the haplotype structure of the human genome ensures that multiple SNPs lie in high linkage disequilibrium (LD) with the peak association signal so that it is rarely possible to promote one variant as causal on statistical evidence alone. Second, it is now clear that at least one-third of loci harbor multiple independent associations, most with overlapping credible intervals [4,5,6]. Third, the majority of the risk loci are located in non-coding regions of genes [7,8], where they exert their function through regulation of gene expression. Tools for predicting the function of such causal variants generally have low predictive value [9,10].

Moderate-to-high throughput methods are needed to prioritize likely causal variants by experimentally monitoring their effects on gene expression [11]. Two broad classes of approaches have been described: massively parallel reporter assays and genome editing. Massively parallel reporter assays a couple of short segments of potentially regulatory DNA to guide barcodes, which are transcribed following transfection into cells or animals. Sequencing approaches allow identification of under- or over-represented barcodes, indicating differential expression due, for example, to polymorphisms. Genome editing approaches now most commonly use CRISPR-Cas9 to introduce short insertions, deletions, and substitutions into targetable regions across the whole genome. RNA sequencing or other functional readouts, such as fluorescence of a reporter gene, can be used to monitor the impact of specific variants. Recent CRISPRi and CRISPRa pooled screening assays utilize catalytically dead/inactivated Cas9 enzymes (dCas9) that bind to but do not cut the target site. These modified Cas9s have their endonuclease activity removed, but they are still able to bind to the target sites where they contribute to inhibition or activation of gene expression via fused effector domains, such as KRAB (CRISPRi) and VP64 (CRISPRa). They have enabled high-throughput screening of genomic elements, influencing transcription [12] and cellular phenotypes [13,14,15,16], with single-cell transcriptome readout. However, the majority of these strategies screen regulatory intervals rather than individual SNPs, so they are not appropriate for fine-mapping causal variants.

Here, we showed the feasibility of gene-centric single-cell clonal analysis, focusing on a handful of genes known to influence the risk of inflammatory bowel disease (IBD) through modulation of gene expression in immune cells. Specifically, we chose to examine four genes with evidence for two independent *cis*-expression Quantitative Trait Loci (eQTL) intervals each, as well as GWAS-significant associations with IBD. The CDGSH iron-sulfur domain 1, *CISD1*, and serologically defined colon cancer antigen 3, *SDCCAG3*, genes are associated with both ulcerative colitis and Crohn’s disease [17,18]. The autocrine motility factor receptor, *AMFR*, encodes a glycosylated transmembrane receptor that is also an E3 ubiquitin ligase, knockdown of which in the acute monocytic leukemia cell line, THP-1, induces cell cycle arrest and apoptosis, indicating a critical role for *AMFR* in cell proliferation [19]. *NFXL1* is one of the most up-regulated genes in IL-4 induced macrophages [20].

We used an experimental strategy for targeted SNP evaluation wherein microdeletions targeting candidate eSNPs were introduced by CRISPR-Cas9 and then isolated as single-cell clones on a uniform genetic background. Although homology-directed repair (HDR) would provide a more precise evaluation of allelic replacement, the low efficiency relative to non-homologous end joining (NHEJ) and expectation that indels might have larger effects led us to use NHEJ in these experiments. We chose the HL60 cell line, a pro-myelocytic lineage, which can be induced to undergo differentiation toward neutrophil- or monocyte-like fate, allowing the evaluation of SNP effects in different cell types. Given the challenges in demonstrating conclusively the impacts of a single causal variant, we discussed sources of experimental variance encountered with this strategy, including batch, clonal, and differentiation effects, and used these to derive realistic power estimates for dissection of causal variants. Comparing these estimates with empirically defined eQTL effect sizes, we concluded that this approach is generally incapable of resolving most regulatory associations to single causal variants.

## 2. Materials and Methods

### 2.1. eGenes, Candidate eSNPs, and Control SNP Selection

The eGenes *CISD1* and *SDCCAG3* were chosen due to the colocalization of eQTL signals and association with inflammatory bowel disease [21]. *NFXL1* and *AMFR* were included as they are essential for myeloid cell differentiation. Candidate eSNPs were selected from one of at least two independent eQTL credible intervals at each locus identified in a multiple eQTL studies using stepwise conditional regression [6] in two large peripheral blood microarray datasets—the Consortium for the Architecture of Gene Expression (CAGE) [22] and Framingham Heart Study (FHS). They were also confirmed to be eQTL in monocytes [23]. It remains possible that they are not actually active in HL60 cells or their derivatives, and our experiments should be interpreted with this in mind. We also evaluated each SNP in the credible interval with Combined Annotation Dependent Depletion (CADD) score [24] and evolutionary probability (EP) [25]. In each credible interval, we chose the SNP with the lowest *p*-value, named as “Top SNP”, SNPs with low evolutionary probabilities (EP) of the minor allele and (or) high CADD scores, named as “Both” and “High CADD”, respectively (Table 1). We also picked SNPs as negative controls with no eQTL signals and in linkage equilibrium with the top SNP, named as “Control”. Conditional eQTL profiles can be visualized using our eQTL Hub shiny browser at http://bloodqtlshiny.biosci.gatech.edu/.

### 2.2. SNP-Targeting and gRNA Screening Design

The chromosomal position of each candidate SNP in reference genome hg19 was obtained from the dbSNP database [27] by searching their RSID. The sequences flanking the targeted SNP were fetched from the NCBI Reference Sequence (RefSeq), providing a gRNA screening window [28]. In each window, all the 19-base sequences followed by the correct *Streptococcus pyogenes* Cas9 protospacer adjacent motif (PAM) sequence (NGG) were collected as candidate gRNAs. gRNAs with GC rate over 80% or less than 10% were filtered out to assure better-cutting performance, and only the gRNAs with a distance of cut site to targeted SNP not more than 10 nucleotides were selected for off-target effect analysis. The in silico predictions of their off-target effects were tested using COSMID [29]. The online tool is available through https://crispr.bme.gatech.edu/.

### 2.3. Single-Cell Clone Generation

HL60 (ATCC, Manassas, VA, USA, CCL-240) and HL60/S4 (ATCC, Manassas, VA, USA, CRL-3306) cells were grown in suspension at 2 × 10^5^ to 1 × 10^6^ cells/mL in RPMI-1640 with 10% FBS, 2 mM L-glutamine, and 100 μg/mL normocin. After culturing for 18 h to 24 h, cells were pelleted at 200 g for 3 min. Used media was collected and filtered to obtain conditioned media. Bulk cell suspensions were serially diluted on a 96-well plate with conditioned media to facilitate cell growth. Statistically, there were wells that only had a single-cell. Alternatively, some single-cell clones were generated by sorting bulk cells by flow cytometry on a BD FacsAria Fusion with 100-micron nozzle at 37 °C and seeded onto each well of a 96-well plate with the same conditioned media.

### 2.4. Myeloid Lineage Differentiation

The differentiation of cells into neutrophils was achieved by culturing with 1 μM retinoic acid (RA) [30]. Cells were seeded 18 h before treatment at 2 × 10^5^ cells/mL. HL60 cells were treated for 4 days, and HL60/S4 were treated for 2 days. During differentiation, cell density and viability were checked every 24 h to maintain 2 × 10^5^ to 1 × 10^6^ cells/mL cell density. Additional culture media with RA was added if needed. Cells treated with the same volume of ethanol were used as a negative control.

Differentiation of cells into monocytes was achieved by culturing with 100 nM α1, 25-dihydroxyvitamin D3 (D3) dissolved in ethanol [31]. Cells were seeded at 1.5 × 10^5^ cells/mL at least 18 h before treatment. Both HL60 cells and HL60/S4 were treated for 3 days. During differentiation, alive cell density was checked and normalized every 24 h to maintain 2.5 × 10^5^/mL cell density. Additional culture media with D3 was added if required. Cells treated with the same volume of ethanol were used as negative controls.

### 2.5. Flow Cytometry

After collection, cells were washed with PBS twice at room temperature. Cells under neutrophil differentiation were then incubated with 7-aminoactinomycin D (7-AAD) (ThermoFisher Scientific, Waltham, MA, USA, cat. No. A1310) and PE-conjugated mouse anti-human CD11b (clone ICRF44) (BD Biosciences, San Jose, CA, USA, cat. No. 557321) or PE-conjugated isotype control mouse mAb (clone: MOPC-21) (Biolegend, San Diego, CA, USA, cat. No. 400112) for 40 min at 4 °C in the dark. Samples were analyzed by BD FacsAria Fusion with a 100-micron nozzle at 4 °C. Cells under monocyte differentiation were incubated with V450 mouse anti-human CD14 (BD Biosciences, San Jose, CA, USA, cat. No. 560349) and adenomatous polyposis coli (APC) mouse anti-human CD71 (BD Biosciences, San Jose, CA, USA, cat. No. 551374) or V450 mouse IgG2b (BD Biosciences, San Jose, CA, USA, cat. No. 560374) and APC mouse IgG1 (BD Biosciences, San Jose, CA, USA, cat. No. 555751) for isotype control. Samples were analyzed by BD FACSMelody at 4 °C. All data were analyzed with FlowJo software v10.6.1 downloaded from https://www.flowjo.com/.

### 2.6. Immunofluorescence

After collection, cells were washed with PBS twice at room temperature. Then, cells were incubated with Hoechst-33342 (ThermoFisher, Waltham, MA, USA, cat. No. H3570) for 10 to 15 min at 37 °C in the dark. Ten microliters of the cell suspension were used to make a slide, which was sealed with clear nail polish. UV excitation and microscopic imaging were done on an Olympus IX73 inverted microscope system.

### 2.7. RNA Isolation

Cells were grown in suspension at 2 × 10^5^ to 1 × 10^6^ cells/mL in RPMI-1640 with 10% FBS, 2 mM L-glutamine, and 100 μg/mL normocin. Cells were seeded at 2 × 10^5^ cells/mL 18 h to 24 h before extraction. Each clone had two biological replicates, except bulk HL60/S4. One million cells from each sample were collected by centrifuging at 300 g for 5 min. Total RNA was isolated and purified by RNeasy Plus Mini Kit (Qiagen, Hilden, Germany, cat. Nos. 74,134 and 74,136). Quality control of RNA samples was assessed with a Bioanalyzer 2100 instrument (Agilent, Santa Clara, CA, USA).

### 2.8. Bulk RNA-Seq and Differential Gene Expression Analysis

cDNA library preparation for single-cell clones was performed using Illumina TruSeq Stranded Sample Preparation, Low Sample (LS) Protocol. Sequencing was performed on an Illumina HiSeq 2500 at Georgia Tech, generating 100 bp paired-end libraries with an average of 51.8 million paired reads per sample. Library preparation for differentiated cells was performed using the NEBNext Ultra II Directional RNA Library Prep Kit for Illumina (New England BioLabs, Ipswich, MA, USA, cat. No. E7760S). Sequencing was performed on Illumina NextSeq, high output, generating 75 bp paired-end libraries with an average of 36 million paired reads per sample. The gene expression data is available at the Gene Expression Omnibus (GEO) under the accession code GSE135507.

RNA-Seq quality control was initiated with Trim Galore, which was used to trim the 13 bp Illumina standard adapter (‘AGATCGGAAGAGC’) by default, after which quality control was reported by FastQC. Reads were mapped to the hg38 human reference genome by STAR [32], and on average, the mapped reads were 90% of total reads. Aligned sequencing reads were counted with the intersection-strict mode in HTSeq [33] to get read counts for each gene. Scale factors of each sample were computed using the trimmed mean of the M-value (TMM) algorithm in the R package, edgeR [34]. Raw read counts were normalized by scale factors and then transformed into log2 counts per million reads (CPM). Genes were kept if expressed in at least three samples. A total of 11,746 genes were kept in single-cell clone RNA-Seq, while 13,485 genes were kept in differentiated cell RNA-Seq.

Differential gene expression analysis was conducted in edgeR with generalized linear models to contrast the effects of each treatment group. Pairwise comparisons between control and neutrophil derivative, control and monocyte derivative, as well as within each clone of each type of cell, were performed. Likelihood ratio tests were assessed to obtain lists of differentially expressed genes and following Benjamini-Hochberg false discovery rate correction.

Gene ontology analysis was performed using ToppFun [35]. By uploading a list of differentially expressed genes (FDR < 0.001) from the differential gene expression analysis into the website, functional enrichment features were listed, including pathways, Gene Ontology (GO) terms, and phenotypes. Gene ontology analysis was also performed by enrichR [36,37], with four sets of differentially expressed genes (FDR < 0.001) uniquely in HL60 monocyte (968 genes), HL60/S4 monocytes (521 genes), HL60 neutrophils (1462 genes), and HL60/S4 neutrophils (2275 genes).

Principal component analysis (PCA) was performed on 17 single-cell clone samples and 47 differentiated cell samples by “prcomp” function in R, with default settings. Principal variance component analysis (PVCA) was performed in JMP Genomics 8 (SAS Institute, Cary, NC, USA), which sums the weighted proportions of each variance component associated with covariates of interest in order to estimate the overall contribution of biological and technical factors to the gene expression variation. Plots were plotted with R package, ggplot2.

### 2.9. Variant Calling

Variants were called by GATK [38,39] best practice RNA-seq short variant discovery (SNPs and Indels). Raw RNA-seq reads were mapped to hg19 by STAR [32]. “SplitNCigarReads” was used to split reads that span introns and hard clip mismatching overhangs. Variants were called by “HaplotypeCaller” with default settings. Due to the high false-positive rate of calling variants from RNA-seq data, the “VariantFiltration” function was used to filter potential false-positive calls. Clusters of at least three SNPs within a window of 35 bases were excluded, and calls with read depth lower than 50 were filtered. Moreover, the variant calls were only included if they were consistent in the two biological replicates of the same clone, and only exonic polymorphisms were counted.

### 2.10. Fluidigm qRT-PCR

Fluidigm real-time qPCR was conducted on a 48 × 48 nanoscale microfluidic chip with 48 EvaGreen probes targeting transcripts of the CRISPR targeted genes, as well as a representative set of lymphoid and myeloid cell marker genes [40], and housekeeping genes. The 48 array samples included single-cell clone CRISPR-edited HL60/S4 from two batches and experimental controls. A total of 2304 qRT-PCR assays with 30 amplification cycles were conducted in parallel according to the manufacturer’s protocol. The average Ct value was computed at the exponential phase of each PCR amplification reaction. Since large Ct values correspond, counter-intuitively, to low expression, modified expression values were computed as the Ct values subtracted from 30 (the maximum number of PCR cycles), and the negative outputs were set as 0. This results in a range from null to 30, where each increment, in theory, represents a doubling of initial transcript abundance. To clean up the data, samples with more than 40 unexpressed genes and probes expressed in less than 5 samples were removed. Processed expression data and sample phenotypic information are provided in Appendix A, respectively. We noted that numerous studies have established the high sensitivity of Fluidigm relative to standard qRT-PCR [41,42,43] and that all expression levels were in the normal range of detection and not subject to drop-out seen with very low abundance transcripts.

### 2.11. Plasmid Construction

The SpyCas9 expressing plasmid pX330-U6-Chimeric_BB-CBh-hSpCas9 [44] (Addgene plasmid #42230) was a gift from Dr. Feng Zhang. The pX330 vector was digested by BbsI. For each designed gRNA sequence, a pair of annealed oligos was cloned into the vector before the gRNA scaffold and after the U6 promoter. All clones were validated by Sanger sequencing (Eurofins Genomics, Louisville, KY, USA).

### 2.12. CRISPR-Edited Single-Cell Clones Generation

A total of 2 × 10^5^ HL60/S4 clone 3 cells and 1 μg of pX330 plasmid per nucleofection reaction (program CA-137, solution SF) were electroporated using the Lonza Nucleofector 4-D based on the manufacturer’s protocol. One microgram of pmaxGFP™ vector per nucleofection reaction was co-transfected as the reporter. The cells were cultured at 37 °C for 72 h after nucleofection, and the GFP-positive cells were sorted individually by BD FACSMelody to make single-cell clones following standard protocols. Post-sorting, cells were grown for a week before harvesting and DNA extraction. DNA was extracted using Quick-DNA Miniprep Plus Kit (Zymo Research, Irvine, CA, USA, cat. No. D3024) following the manufacturer’s protocol. For each target locus, a PCR product was amplified from the genomic DNA of cells modified by CRISPR-Cas9 and analyzed by Sanger sequencing (Eurofins Genomics, Louisville, KY, USA). The genotype of clones selected in this study is shown in Appendix A, and the number of clones screened and the mutations observed per clone are shown in Appendix A.

### 2.13. Power Simulation Studies

Power analysis was performed using the mixed model power expression utility in JMP Genomics (SAS Institute, Cary, NC, USA). We created a design file with duplicates of 10 guide RNAs and designated one guide as the causal variant. Additional random effect options for representing batch effects (distributing the guides across into two batches of 5) and clone effects (where the causal variant was represented by two different clones) allowed modeling of the impact of these additional sources of variance. We assessed power at α = 0.05, 0.01, and 0.001 for effect sizes of the causal variant in increments of 0.1 standard deviation units (sdu) between 0 and 2, assuming experiments with 2, 4, 8, or 16 replicates of each guide. Batch and clone effects were assumed to be 0.1 or 0.2 sdu. For additional analysis, three of the guides were assumed to affect gene expression, modeling the situation where multiple linked variants account for an eQTL effect.

## 3. Results

### 3.1. Effect of Clonal Variability on Gene Expression in HL60 Cells

Since genetic screens are best performed in uniform genetic backgrounds under conditions where environmental variation can be carefully controlled, we started by evaluating the magnitude of the effect of biological and technical factors on gene expression in HL60 cells. HL60 is a pro-myeloid cell line derived from a person with acute promyelocytic leukemia [45,46]. It is known to be homozygous for a *TP53* deletion and a *CDKN2A* premature stop codon and heterozygous for an *NRAS* missense substitution. The main factors of interest were (i) batch effects, (ii) HL60 sub-type, (iii) clonal heterogeneity, and (iv) differentiation status. A derivative known as HL60/S4 has been isolated, which is reported to more efficiently differentiate into myeloid derivatives, such as neutrophils and macrophages [47]. Given the almost 40 years in culture, we reasoned that point mutations that are likely to affect overall gene expression might have accumulated, and, to control this, we isolated three single-cell clones (labeled 1 through 3) of HL60 and four single-cell clones (labeled a through d) of HL60/S4. Differences in growth rates among clones and relative to the bulk parental line were noted.

Clonal variability in gene expression was monitored by bulk RNA-seq of two batches for each of the seven single-cell clones and two parental lines. Figure 1a plots the first two principal components (PC) of expression of 11,746 expressed genes detected with an average depth of over 50 million paired-end 100 bp reads per sample. PC1 separated the two HL60 sub-types unambiguously, and 85% of the variance attributable to the first five PC (86.8% of total variance) was between HL60 and HL60/S4 cells. Individual clones separated along PC2 with relatively little separation between replicates, with the parental lines taking intermediate values. Just 14% of the variance was among clones, but residual replicate effects accounted for less than 1% of it (Figure 1b). These results confirmed that single-cell clones were likely genetically differentiated, implying that, as far as possible, CRISPR-Cas9 editing should be performed on a purified clone.

The extent of genetic differentiation of single-cell clones was evaluated by calling genotypes directly from the RNA-seq data. Given that false-positive calls are elevated due to errors induced by the reverse transcriptase during cDNA preparation, and that allele-specific expression causes SNP ratios not observed in genomic DNA sequence data, we applied variant hard filtering in GATK. Clusters of at least three SNPs within a window of 35 bases were excluded, the variant calls were only included if they were consistent in the two biological replicates of the same clone, and only exonic polymorphisms were counted. On average, each of the HL60 single-cell clones differed from the bulk consensus sequence at 103 of the 7482 single nucleotide variants (SNVs) (1.38%), passing our hard filters. A little over fifty percent more divergence and 166 of 7104 SNVs (2.34%) were uniquely observed in HL60/S4 pairwise clonal comparisons with the bulk HL60/S4 consensus. Furthermore, approximately 3% of the total SNVs were different in the comparison of bulk HL60/S4 and HL60 lines and their derivatives, indicating that there was considerable genetic variability both between the two lines and in single-cell clones. Similar findings have been reported [48] in an analysis of somatic mutation accumulation in a cancer cell line.

Next, we asked how consistent chemical-induced differentiation is across clones. Each of the single-cell clones, with the exception of HL60/S4 clone d, was treated with 1 μM retinoic acid for 4 days (HL60) or 2 days (HL60/S4) in order to generate neutrophil-like cells or with 100 nM α1,25-dihydroxyvitamin D3 for 3 days in order to generate monocyte-like cells. Appendix A shows characteristics of the cells stained with Hoechst to monitor changes in the morphology of the nucleus, 7-AAD to monitor cell viability, and CD11b, a neutrophil marker. Growth conditions were chosen to optimize the balance of cell differentiation and viability, which also varied among clones. As previously reported [47], HL60/S4 cells more readily differentiated toward neutrophil fate than did HL60 cells. Appendix A confirms initiation of CD14 expression, as well as the loss of CD71, both markers of monocyte fate, to similar degrees in both bulk HL60 and HL60/S4, though variation among clones of HL60 was also seen (also Appendix A), including variability of cell surface marker expression at baseline.

As with the untreated clones, gene expression was again observed to vary substantially between the two sub-types and among clones, with a generally uniform response to treatment and relatively small differences between replicates (Figure 1c). In a joint analysis, HL60/S4 cells tended to have more positive values of PC1 and negative values of PC2 than HL60, and the overall cell-type accounted for 38.5% of the variance captured by the first five PC (83.9% of total variance). Neutrophils occupied an intermediate position between monocytes and undifferentiated cells along both PC axes, and cell fate captured 36.8% of the variance. At baseline, HL60/S4 cells appeared to be more divergent from the derived neutrophil-like and, especially, monocyte-like cells than were HL60 from their derivatives. Clonal differences remained significantly higher than replicate effects.

In total, 5885 and 3319 genes (FDR < 0.0001) were identified that were differentially expressed before and after monocyte and neutrophil lineage differentiation across all clones of two cell types—HL60 and HL60/S4—respectively.

After differentiation, HL60/S4-derived monocyte cells were more transcriptionally divergent from their parental cells than were HL60-derived monocytes: 7381 monocytic differentially expressed genes were detected in HL60/S4, compared with 4167 genes in HL60. *B2M*, a neutrophil-specific differentiation marker, was one of the 4167 genes that were differentially expressed in the neutrophil-derived clone a, clone b, and HL60 bulk cells. There were 5079 differentially expressed genes in the monocyte derivatives of HL60, including the transcription factors *CEBPE*, specifically in clone c derivatives, and *PU.1* in clone b derivatives. Similar gene markers were also documented in a time course of myeloid differentiation [45], although we observed a higher number of differentially expressed genes at the terminal differentiated stage of monocytes than neutrophils, whereas the opposite pattern was found at 6 h post-differentiation [49].

Differences in the degree of inter-clonal differentiation were also detected (Appendix A). For the monocyte derivatives, 1781 genes were differentially expressed relative to undifferentiated cells in all of the clones of the two cell types, and these were enriched in cell cycle, neutrophil degranulation, and rRNA processing pathways. On the other hand, 968 genes were uniquely differentially expressed in the HL60 clonal comparisons, also showing enrichment for neutrophil degranulation and innate immune system pathways. Gene ontology (GO) and pathway analysis was performed by Toppfun, and the significant GO terms and pathways (Bonferroni corrected *p*-value < 0.00001) for these 968 genes are listed in Appendix A. Similarly, for neutrophil lineage differentiation, 413 differentially expressed genes were shared by HL60 and HL60/S4, enriched for neutrophil degranulation, innate immune system activity, interleukin-10 signaling, chemokine signaling, and cytokine signaling pathways. There were 1462 and 2275 clonal-specific differentially expressed genes in HL60 clones and HL60/S4 clones, respectively, engaging pathways involved in cell cycle and mitochondrial function, and translation and rRNA processing were also enriched. Significant GO terms and pathways (Bonferroni corrected *p*-value < 0.00001) for HL60 and HL60/S4 are shown in Appendix A, respectively. Gene ontology enrichment analysis of uniquely differentially expressed genes was also performed using the gene set enrichment tool Enrichr [36,37], with results summarized in Appendix A.

Taken together these results implied that single-cell clones differ in basal gene expression, and although they respond similarly to treatment with retinoic acid or vitamin D3, clonal differences need to be accounted for when evaluating the effect of CRISPR-Cas9 mutagenesis of regulatory regions of target genes.

### 3.2. Isolation and Evaluation of CRISPR-Edited Single-Cell Clones

We selected seven SNPs in four genes for our initial evaluation of the effect of NHEJ-based CRISPR mutagenesis in HL60/S4 clone 3 as a uniform genetic background. *SDCCAG3*, *NFXL1*, and *AMFR* were each targeted for a single peak eQTL SNP detected by whole blood gene expression, whereas *CISD1* was targeted with four SNPs in one credible eQTL interval. Potential off-target sites of each gRNA with up to two mismatches are provided in Appendix A. With genome-wide bioinformatic screening, none of the potential off-target sites were located in coding regions, and the gRNAs had no extra perfect match other than the designed target site. Bulk transfection efficiency was 24.8% based on the percentage of cells expressing GFP signal. GFP-positive cells were considered capable of uptaking plasmid vectors and were single-cell sorted to enrich the edited cells. Of all expanded GFP-positive single-cell clones, 23 out of 166 had obtained Indels, eight of which had removed the target SNP at both allelic copies, while the remainder affected sequences immediately adjacent to the target SNP or only had SNP removal in one allele.

RNA-seq would be prohibitively expensive for comparing gene expression on the scale of dozens of multiple replicated clones, so we next evaluated the potential of high throughput nanoscale quantitative RT-PCR to detect subtle differences in transcript abundance. A 48 × 48 Fluidigm chip was designed, facilitating the measurement of 48 genes (including the four targets, housekeeping controls, and various markers of expression in diverse immune cell type) in 48 samples. The HL60/S4 parental cell line and eight clones were chosen for profiling, one for each guide RNA, and each was grown in duplicate in suspension for 18–24 h, with half the sample frozen down for storage, and the other half used for RNA preparation from fresh cells.

For ease of interpretation, we subtracted the Ct value for each measurement from the number of PCR cycles, 30, resulting in expression values where high values corresponded to high expression. Figure 2a shows that this resulted in a bimodal distribution of gene expression measures, with the smaller peak representing low-abundance transcripts. There was a major difference in the profiles of the frozen and fresh cells, accounting for almost two-thirds of the variance explained by the first five PCs (99.1%) (Figure 2b). To correct for this batch effect, we used Combat, which also standardized the data to a mean of zero and standard deviation of one (Figure 2c). On this scale, most of the variance was now among samples, whereas 9% of first five PCs (99.1%) distinguished clones by which gene was targeted, and 9% was due to differences among gRNAs for *CISD1* (Figure 2d). This implied either that single-gene knockouts affected the expression of a substantial number of other genes, in each clone, or that there was substantial variability among clones that by chance correlated with the nature of the guide RNA. We also observed that normalized *CISD1* expression was lower in cells edited by each of the four gRNAs targeting *CISD1* than in the untreated control parental cell line (Figure 2e). Clone RG17 affected a control SNP in high LD with the peak eQTL but with low CADD score [24,50] and high evolutionary probability [25] of the alternate allele and was the only clone not significantly different from the parental line. However, since it is unlikely that each of the other three sites causally influences gene expression, this result served as a further caution that the process of transfection with CRISPR reagents itself might influence cell growth and gene activity.

Similarly, inconsistent results were obtained for the other three genes, as summarized in Figure 3 and Appendix A. Each panel shows box-and-whisker plots for each of the seven guide RNAs and control HL60/S4 cells, with the mean and interquartile range of nine single-cell clones measured with two different PCR probes for three of the genes and one for *SDCCAG3*. In no case was the expression the most extreme for the guide RNA corresponding to the linked gene. For example, *AMFR* expression was highest in cells carrying a mutation in the RG16 guide, disrupting a candidate regulatory site in *CISD1*, whereas *AMFR* expression itself was, on average, the closest to expression in the control cells. Disregarding the control, there were also no cases where the appropriate guide RNA was significantly different from the remaining guides. These results implied either that the selected SNPs were not causal or that the effect sizes of causal variants were too small relative to the observed experimental variability to detect differential expression.

### 3.3. Simulation Studies to Establish Power of Fluidigm-Based Single-Cell Regulatory Assessment

We used these results to guide our design and interpretation of power calculations for experiments designed to determine the effect of single regulatory site disruption. Our baseline scenario assumed targeting of 10 polymorphisms in a single credible interval in which a single eQTL was assumed to account for at least 10% of the variance in transcript abundance at the locus. Such an eQTL corresponded to a difference of approximately 1 standard deviation unit (sdu) in a quantitative assay, such as Fluidigm qRT-PCR or RNA-seq. Given that most single-cell CRISPR-edited clones are heterozygous, it also corresponded to a substitution effect whereby the mutant allele increased or decreased the measured transcript by 1 sdu. We used the mixed model power calculator in JMP-Genomics (Cary, NC) to evaluate the sample size needed to detect an effect of this magnitude, given varying levels of clonal variation, batch effects, and mutation differences.

For the baseline scenario, where there are neither batch nor clonal effects, 80% power to demonstrate that one SNP had an effect that was at least 1 sdu different from the other nine SNPs was achieved with eight replicates of each of the ten clones (Figure 4a,f). Sixteen replicates would enable detection of an effect as small as 0.7 sdu, but four replicates would only be powered to detect a substitution effect of 1.5 sdu. However, the experimental data indicated that individual clones generally did vary, as a consequence of genetic background effects if the transfected cell line was not isogenic, or due to growth differences among aliquots. Modeling these differences as a random effect of just 0.2 sdu among the ten clones demonstrated a dramatic reduction in power to detect the main effect (Figure 4b,g). With eight replicates, only an effect size of 1.7 sdu was reliably detected, though 40% power was still obtained for an effect size of 1 sdu. Doubling the size of the experiment only slightly improved the power, whereas four replicates only facilitated the detection of effect sizes of 2 sdu. If we further considered the scenario with a batch effect whereby half the clones had an additional random effect of 0.2 sdu (perhaps because they were grown at a different time), then power reduced yet again, as expected (Figure 4c,h).

A perhaps more realistic scenario is where different edits of the same polymorphic site also have different impacts on gene expression. This could either be because the precise nature of the deletion matters or because the independent clones have slightly different growth properties. We modeled this scenario by allowing for two different clones representing the causal variant, also with a 0.2 sdu random effect difference, the same as the effect of the other nine guide RNAs. In this case (Figure 4d,i), 80% power was never achieved, so it would take greater levels of replication, at least, of the putative causal variant to see a substitution effect in the range of 1 sdu.

A related situation was where more than one of the polymorphisms in the credible interval was responsible for the eQTL effect—for example, three sites in high LD might each account for 0.33 sdu, summing to a combined effect of 1 sdu. To model this, we set three of the guide RNAs to be causal, with the other seven non-functional but retained 0.2 sdu differences among clones. Figure 4e,j show that power was greater than the same scenario with one causal variant and approximately the same as with one causal variant and no differences among the remaining clones. Power was actually greater with fewer replicates (red and blue curves), but, with eight replicates, 80% power still only detected an effect size of 1 sdu, which was three times larger than the presumed individual effect sizes of the contributing causal variants.

## 4. Discussion

Multiple studies have recently reported good success in mapping regulatory intervals using high throughput approaches in human cells. A previous study [51] scanned across over 100 kb of regulatory DNA in the *TP53* and *ESR1* genes using positive selection for proliferation to enrich cells with aberrantly low expression of the target transcription factors, defining several intervals enriched for signals that overlap with transcription factor binding sites. This approach is, however, dependent on the ability to select on the locus, and similar to methods that sort on the basis of an engineered selectable fluorescence protein [52], only identifies high-impact sites without necessarily discriminating effects of polymorphic sites. Another approach [53] used CRISPRa to map enhancer elements by virtue of activation of regulatory protein-DNA interactions, filtering a handful of short DNA stretches from hundreds of kb of intergenic sequence in the *IL-2RA* gene, but again without the ability to resolve which of the SNPs in a credible interval are responsible for an eQTL. Expression CROP-seq is powered to fine-map eSNPs with 10%–20% effect size within credible intervals by characterizing hundreds of CRISPR/Cas9 genetically mutated single-cell transcriptomes in parallel [54]. Tewhey et al. first demonstrated the utility of massively parallel reporter assays, including the ability to discriminate between alleles at a pre-defined site [55]. Their results and findings from others [56,57] implied that at least 5% of all polymorphisms in regulatory DNA had the potential to regulate target gene expression. The concern remains though that such effects may be artifacts of short reporter genes assayed outside the context of chromatin and complex regulatory interactions.

Our approach instead borrows from classical quantitative genetic screens in model organisms, such as Drosophila and yeast. The objective was to create a panel of genetic perturbations in an isogenic background, evaluating the quantitative impact of each variant relative to the frequency distribution of effects of all other perturbations. For example, *p*-element insertion screens cleanly identified dozens of genes, influencing aging, bristle number, and aspects of fly behavior [58,59]. Closer to our experiments, another study [60] engineered a tiling path across the regulatory region of the *TDH3* gene in *Saccharomyces cerevisiae* and used flow cytometry to quantify gene expression of hundreds of strains, drawing inferences about the impact of stabilizing selection on transcription. We reasoned that a similar approach should be powerful for moderate-sized laboratories without extensive experience in human cell culture. Even though we, and others, have successfully documented regulatory effects of CRISPR-Cas9-mutagenized candidate mutations of large effect [61,62], the results here applied to typical moderate-effect size eQTL do not support this as a general protocol. The remainder of the discussion deals with multiple constraints on the effectiveness of single-cell clone-based screening to dissect credible regulatory intervals in human cell lines.

The first constraint is variability in the mutability of targeted regulatory sites. Our approach was mainly limited in three ways: the requirement of nearby PAM sequences and the short distance between the cut site and targeted SNP, the variable efficiency of different gRNAs, and the distinct Indel pattern for each SNP-targeted gRNA. We started with a list of 250 candidate polymorphisms, approximately 10 each in two independent eQTL intervals of 13 genes, but discovered that only two-thirds of these were suitable CRISPR targets, either because there was no nearby PAM sequence or the target was in repetitive DNA for which it was not possible to design a guide RNA with a unique target sequence. Up to 20% of the remaining sites were predicted to have high probability off-target sites elsewhere in the genome, which might not matter for a scan of *cis*-acting effects but was not ideal. Subsequently, we chose 10 sites as a pilot and screened an average of 24 single-cell clones for each site (23.9 ± 6.7) by Sanger sequencing of the targeted region. As shown in Appendix A, the pilot group had an average of four clones, each with Indels on both alleles (3.8 ± 1.8). The ratio of clones with Indels on both alleles varied from 0% (RG11) to 25% (RG16) so that the theoretical maximum SNP removal rate was different in each gRNA-treated group. RG14, 17, 19, 20, and 34 all had designed cut <5 bp to the targeted SNP, but their percentage of SNP removal on both alleles varied from 0% to 16%, which could be due to variations in the size of Indel mutations, as previously observed [63]. That is to say, many of the CRISPR-induced mutations removed or inserted one or a few nucleotides either side of the polymorphic site without disrupting the polymorphism itself. We concluded that obtaining at least four different clones for a minimum of 20 sites associated with a credible eQTL interval would typically require screening of 500 clones following various iterations of guide RNA design, with less than 100% success and at considerable expense. Allelic replacement by CRISPR-mediated homologous repair would be even more difficult. There are more potential optimizations that may help researchers deal with this constraint. Further optimization can be done in transfection, such as the co-transfection ratio of two plasmids. It is possible that different cell lines would have higher efficiency of mutagenesis. Other CRISPR/Cas9 delivery methods, such as lentivirus transduction, can also be beneficial for more efficient screening.

The second constraint is clonal variability. We started by addressing a major concern with human cell lines, which is the mutational accumulation in culture. Previous studies [48] showed that tumor cell lines diverge genetically in as few as a dozen passages, resulting in divergent drug responses and gene expression profiles. Accordingly, single-cell cultures of HL60 and the derivative HL60/S4 cell lines are different at the DNA sequence level and have significantly different transcriptomes, both with and without chemical stimulation of differentiation. For a considerable proportion of genes, these differences are of a similar order of magnitude as expected eQTL effects, namely, 20% to 50% differences in normalized abundance. While this observation strongly supports the decision to mutagenize a single-cell clone, genetic differences may not actually be the major source of clonal variation. Mammalian, including human, cells are much more difficult to culture than yeast or bacteria, as thawed aliquots of frozen lines are well known to differ in growth rates and viability. The technical replicates in Figure 1 were all grown in parallel, so did not capture this type of batch effect, which we had not sought to quantify. However, we noted that the parallel culture of the nine mutant clones analyzed was made difficult by variable growth rates and that some thaws failed to grow at all, requiring the expansion of new aliquots. Consequently, batch effects of single-cell clones are a hidden but likely considerable source of gene expression variability.

A third constraint is an expense. Assuming that the cost of RNA sequencing, including cell culture, RNA preparation, library construction, and quality control, could be reduced to $100 a sample using, for example, 3′ tagging, an experiment with eight replicates of 20 clones would still cost $16,000. Instead, we adopted a nanoscale quantitative RT-PCR approach, the 48 × 48 Fluidigm array. Each of the data points in Figure 3 was actually the average of four technical replicate qRT-PCR reactions on one plate at the cost of just $1.20 per assay (not including culture and RNA preparation). Technical repeatability is very high with repeated measures typically within 10%, also allowing measurement of dozens of genes simultaneously, so Fluidigm, or similar methods like Nanostring, provides a feasible approach in theory.

However, the fourth constraint, statistical power, emerged as the most serious impediment. A typical eQTL explains between 10% and 20% of the variance in expression of the gene it influences, which corresponds approximately to each allele increasing or decreasing transcript abundance between 0.5 and 1 standard deviation units. We modeled the power to detect such an effect in 80% of experiments, given the variance components observed in our experiments, and found that in the best-case scenario, eight biological replicates would be needed to reliably detect a 1 sdu effect. However, with the addition of modest batch effects, subtle guide RNA differences within a locus, and small differences between different mutations induced by the same clone, power dropped considerably. All such effects are apparent in Figure 3, suggesting that the single clone analyses, while demonstrably capable of discriminating very large regulatory effects of 2 or more sdu, are not generally likely to be detected with this approach. It is possible that cell lines other than HL60 may provide more repeatable results than those described here, which may improve power under some circumstances. In this sense, independent valuation of the magnitude of batch effects for different cell lines under different growth conditions may be advisable, though we doubt that it will make single-cell mutagenesis an optimal screening approach.

Finally, a fifth constraint is an assumption that each eQTL can be reduced to a single eSNP. This is the parsimonious assumption and fits readily with the conception that regulatory SNPs exert their effects by altering the binding affinity for a specific transcription factor. Even though most eQTL span 100 or more polymorphisms in a credible interval, the general assumption is that prioritizing variants according to functional criteria and evolutionary conservation, using scores, such as CADD or LINSIGHT, reduces the search space to fewer than ten candidates. However, given that these variants are in tight linkage disequilibrium with similar frequencies [10], if they have similar functional scores, then it is possible that the observed univariate eQTL effect is actually due to the summation of two or smaller contributing effects. Under this scenario, the power to detect multiple causal variants is also reduced.

These considerations and the overwhelmingly negative results of our experiments lead us to the recommendation not to pursue single clone-based profiling as a general approach to the fine mapping of regulatory variants. Despite the conceptual limitation that effects are evaluated outside the context of normal chromatin, massively parallel reporter assays seem to be more powerful and subject to less experimental constraint.

## Figures and Tables

**Figure 1 genes-11-00504-f001:**
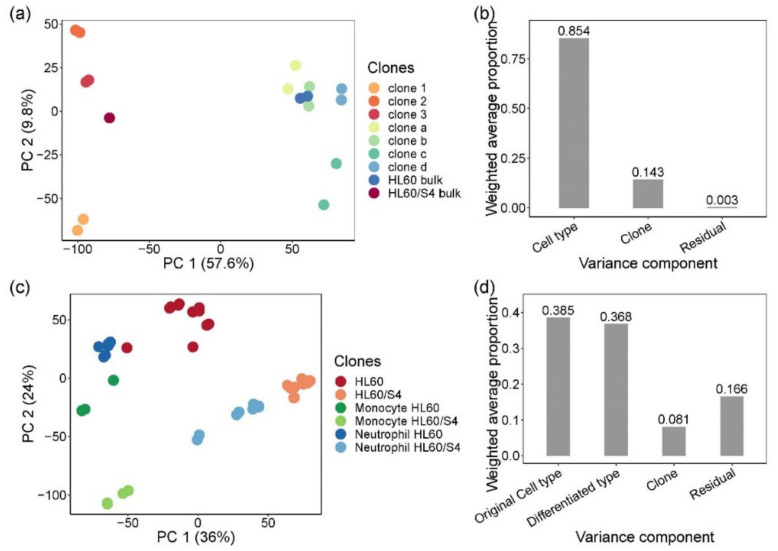
Heterogeneity of gene expression in single-cell clones and myeloid lineage differentiated clones. (**a**) Principal component analysis (PCA) of bulk RNA sequencing of parental single-cell clones and bulk cells. PCA was performed on a normalized log2 CPM count expression matrix of 17 samples from HL60- and HL60/S4-generated single-cell clones. Each dot represents 17 samples, two biological replicates for each clone and bulk, except for HL60/S4 bulk. Samples are colored by clones: warm color dots are samples from HL60/S4 cell lines, while cold color dots are samples from HL60 cell lines. PC1 separated samples by cell type, explaining 57.6% of the total variation. PC2 separated samples by clones, representing 9.8% of the total variation. (**b**) Principal variance component analysis showed the weighted average proportion of each variance component—cell type (85.4%), clone (14.3%), and residual (0.3%)—all of which explained variance captured by the first five principal components (86.8% of total variance). The majority of the total expression variance of single-cell clones was explained by cell type and clone variance components. (**c**) Principal component analysis of bulk RNA sequencing of myeloid lineage differentiated clones, performed by normalized log2 counts per million (CPM) expression matrix. Each dot represents 47 samples from differentiated monocytes and neutrophils and undifferentiated control cells, two biological replicates for each stimulation on each clone. Clone d was excluded due to sequencing error. Samples are colored by cell type and differentiation lineages: monocytes are green, neutrophils are blue, and control cells are red. To distinguish the original cell type of each sample, HL60 cells are dark colors, and HL60/S4 cells are light colors. (**d**) Principal variance component analysis showed the weighted average proportion of each variance component—original cell type (38.5%), differentiated type (36.8%), clone (8.1%), and residual (16.6%)—all of which explained variance captured by the first five principal components (83.9% of total variance). The 16.6% of unexplained variance might be from the variance of biological replicates and cultural differences between two labs.

**Figure 2 genes-11-00504-f002:**
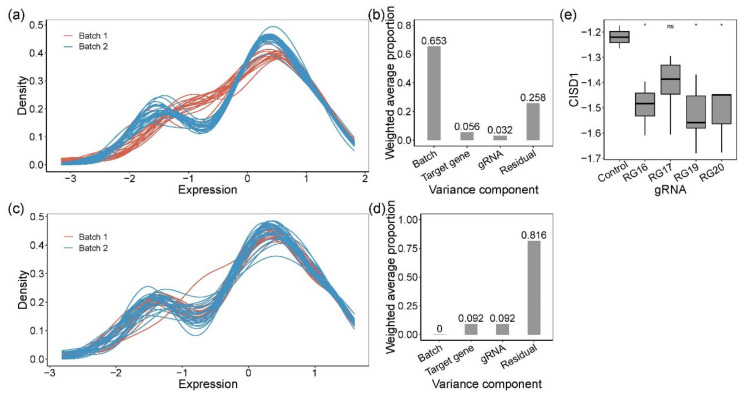
Quantification of gene expression by Fluidigm qRT-PCR and analysis of the variance components. Kernel density plot of standardized gene expression from each sample, color-coded by batches, before (**a**) and after (**c**) removing batch effect. Before (**b**) and after (**d**) batch effect correction, principal variance component analysis showed the weighted average proportion of each variance component: batch 65.3%, 0%, respectively; target gene 5.6%, 9.2%, respectively; gRNA 3.2%, 9.2%, respectively; residual 25.8%, 81.6%, respectively. All of the components explained variance captured by the first five principal components (99.1% and 99.1% of the total variance, respectively). (**e**) Expression of *CISD1*. Pairwise *t*-tests were used to evaluate the difference between CRISPR-Cas9-edited samples (RG16, RG17, RG19, and RG20) and negative controls. RG16, RG19, and RG20 were significantly different from the negative control. * denotes *p*-value < 0.05; ns, not significant.

**Figure 3 genes-11-00504-f003:**
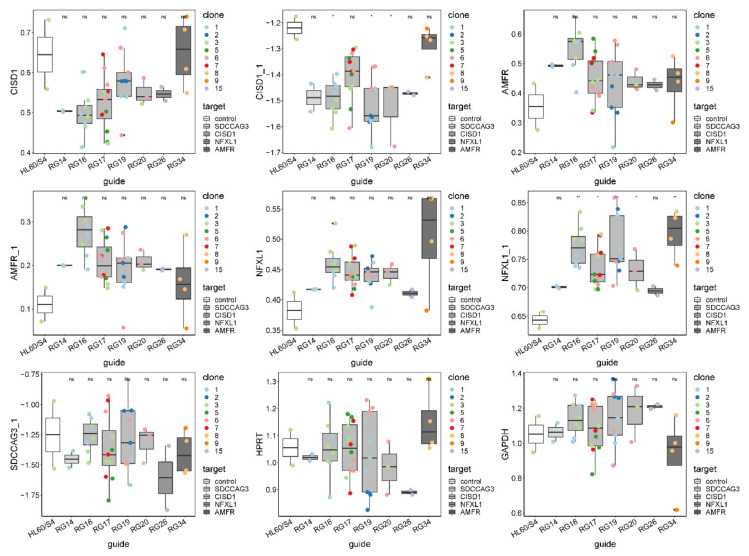
Quantification of all targeted gene expression in all CRISPR-Cas9-edited single-cell clones by Fluidigm qRT-PCR (Appendix A). *HPRT* and *GAPDH* are housekeeping controls. Single-cell clones were grouped by guide RNA, and the expression of seven probes is shown as boxplot across all clones within each guide RNA group. Clones with the same genotype in each guide RNA group are colored-coded. A pairwise t-test was done to test the difference between CRISPR-Cas9-edited clones and HL60/S4 negative control. * denotes *p*-value < 0.05; ns, not significant.

**Figure 4 genes-11-00504-f004:**
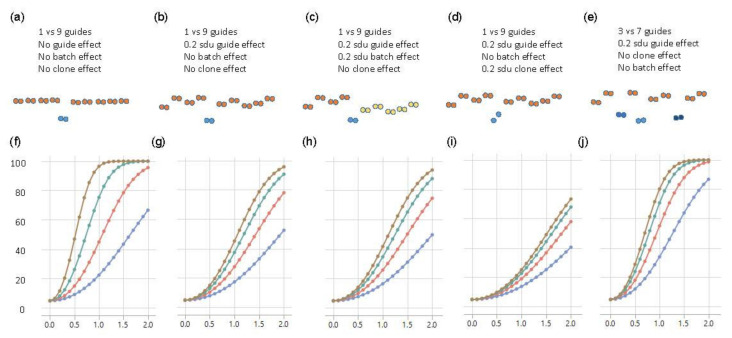
Power curves of Fluidigm-based single-cell clone regulatory assessment of simulation studies. (**a**–**e**) diagrams of five different scenarios and the corresponding panels (**f**–**j**) show the power calculations for exceeding a nominal *p*-value of 0.05, with blue, red, green, and brown curves representing 2, 4, 8, and 16 technical replicates of each clone, respectively. The *y*-axis is the power from 0 to 100 percent, and the *x*-axis is the effect size of eQTL in the standard deviation unit.

**Table 1 genes-11-00504-t001:** Guide RNAs and target SNPs. Each guide RNA targets on the “SNP”, which is within a credible set of “gene”. The effect size (z-score) of each SNP from the eQTLGen browser [26]. “Top SNP” is the SNP with the lowest *p*-value in the credible set. Several criteria were used to predict the likelihood of candidate SNPs: “High CADD” is the SNP with high CADD (Combined Annotation Dependent Depletion) score that has a high level of deleteriousness of its variants, including Indel variants; “Top” is the SNP with the strongest signal of eQTL-mapping; “Both” is the SNP with both high CADD score and low evolutionary probabilities (EP) of the minor allele; “Control” is the negative control SNP in high linkage disequilibrium (LD) with the top SNP but low CADD and normal EP.

gRNA	Gene	Top SNP	SNP	Z-Score	Type	Genome Location	Coding Region
RG14	*SDCCAG3*	rs10870171	rs3812594	−34.60	High CADD	Exon of *SEC16A*	Yes
RG16	*CISD1*	rs4397793	rs4397793	−23.84	Top	Intron of *TFAM*	No
RG17	*CISD1*	rs4397793	rs648138	−70.54	Control	Intergenic of *TFAM*	No
RG19	*CISD1*	rs2590375	rs2590363	−100.37	Both	Intron of *IPMK*	No
RG20	*CISD1*	rs2590375	rs1416763	−100.27	Both	Intron of *CISD1*	No
RG26	*NFXL1*	rs116521751	rs321622	−63.35	Both	Intron of *NIPAL1*	No
RG34	*AMFR*	rs8060037	rs8060037	−14.09	Top	Intron of *NUDT21*	No

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
