# Peer review of "Pitfalls in Single Clone CRISPR-Cas9 Mutagenesis to Fine-Map Regulatory Intervals"

_genes, 2020, doi:10.3390/genes11050504_

Round 1
Reviewer 1 Report
The work entitled “Pitfalls in Single Clone CRISPR-Cas9 Mutagenesis to Fine-map Regulatory Intervals” by Ruoyu Tian, Yidan Pan et al. deals with a very interesting topic, which is the direct evaluation of SNP effects in order to find causal variants. The paper is well-written, well-focused and with a high-quality research design. The technical strategy they propose is a moderate-throughput CRISPR-Cas9 mutagenesis in order to introduce microdeletions across a bunch of candidate eSNPs and then evaluate their effects on expression levels. However, they have found some challenges in the process, thus making their primary aim unfeasible. Nevertheless, even if I am not an expert of reporter assays and genome editing, I think this paper could be of great help for those approaching this field of study. In fact, they have done a massive experimental work highlighting all the possible experimental challenges and pitfalls, thus making this paper a source of technical information and guidelines.
However, it could be worth it explaining better why the authors selected those specific genes and SNPs, as well as give us much more information about the effect sizes of those eSNPs.
Reviewer 2 Report
This study by Tian et al reports data on an attempt to finemap disease conferring variants by CRISPR-Cas9 induced microdeletions targeting candidate eSNPs. During the experiments and data evaluation, the authors encountered multiple problems based on which they conclude that the approach applied in this study has serious limitations. Besides the experimentally identified hurdles, the authors support their conclusion with theoretical considerations in the last section. Despite the negative results, the manuscript is valuable and it may well be of interest of researchers planning to try a similar approach in the future. The text is easy-to-follow, the figures are demonstrative and the reasoning of the authors is clearly explained.
Comments:
- The main weakness is the application of a single cell line (HL60) and its derivatives in the study. If the authors do not want to perform more experiments, they should put more emphasis on this limitation in the manuscript.
- Supplementary tables/figures are not precisely ordered based on citation in the text.
- A couple of typos in the manuscript, e.g. mnocyts (line 178), multiply (line 366), has (been) one (line 381) and [49] (line 481).
Reviewer 3 Report
This paper addresses the challenge to fine-map eQTL SNPs. Although GWAS analyses have identified tens of thousands of loci associated with disease risk, most of the loci contain more than 100 SNPs. There are growing efforts identifying the specific SNPs causally associated with disease risk. With limitations in current approaches, the authors decided to develop a cost-effective single-cell clonal analysis to test the effects of selected SNPs on gene expression. They carefully analyzed the variance of gene expressions across different cell clones, and attributed them to biological and technical reasons. They then perturbed the selected SNPs using CRISPR-Cas9 in cells with isogenic backgrounds. Unfortunately, their experimental results didn’t detect any SNP that had causal effects on associated gene expressions. They interpreted that either the SNPs they selected don’t have the effects indeed, or their experimental system is not sensitive enough to detect moderate effects. They then did simulation studies to predict the power of their experimental system to capture causal SNPs. The conclusion is the single-cell clonal analysis is not recommended as an approach for fine-mapping.
Strength
The authors have done a careful work characterizing the variance of gene expressions across experimental batches, cell sub-types, cell clones, and differentiation status, which is valuable work. They also did simulations to estimate the number of replicates needed to detect moderate levels of affected gene expressions after perturbation with statistical significance.
Weakness
Although the authors made a huge effort trying to capture the alteration in gene expression caused by disruption of selected SNPs, their experimental design doesn’t seem to serve the purpose well.
First of all, the approach they used to disrupt SNP is NHEJ-based CRISPR mutagenesis, which always creates indels flanking cutting site. The resulting change in gene expression, if any, reflects the effects of indels, not SNPs, to be precise. The HDR-based CRISPR or base editors, instead, seem to be more appropriate tools for SNP validation.
Second, is eQTL cell-type-specific? The eQTL SNPs selected by the authors were identified in studies in peripheral blood, while the CRISPR mutagenesis in this paper was performed in HL60/S4 cell line, a pro-myelocytic lineage. Is it possible that the SNPs become non-causal in HL60/S4?
Third, is there any evidence that in the selected eQTLs, the causality can be reduced to a single SNP? How likely it would happen if one estimates?
Fourth, in the paper the authors mentioned that the nucleofection step could have affected gene expression in cells, resulting in similar expression changes across all clones. This possibility could have been tested by setting up mock samples when performing nucleofection. The mock samples should be treated the same way as experimental samples, without adding the gRNA-Cas9 plasmid, or with adding a non-targeting gRNA-Cas9 plasmid.
Fifth, there doesn’t seem to be an evaluation of the detection ability of the Fluidigm qRT-PCR system, making readers wondering how well/sensitive/reproducible the method is in detecting differential expressions. If the authors could characterize to what extent Fluidigm qRT-PCR can detect difference in expression levels with some controls, and show how house-keeping gene expression change across cell clones, their conclusion would be more convincing.
Last, the purpose of this study is to develop an affordable yet rigorous approach for SNP validation. The simulation in section III suggests the single-cell clonal analysis in section II is not feasible. It seems the simulation can be done without the experimental results in section II. If that’s true, are the experiments in section II necessary?
As the authors stated in the paper, considering the large number of single cell clones to be established and many replicates of qRT-PCRs, their method would become difficult both in terms of efforts and expense. Will a HDR-CRISPR/base-editing followed by a single-cell RNA seq work better?
Specific points
Line 54: the description of modified Cas9 “bind to but do not cut the target site” is vague. What do they do?
Line 88: “linkage equilibrium” should be linkage disequilibrium
Line 106-107: confusing
Line 317-318: “At baseline, HL60/S4 cells appear to be more monocyte-like and less neutrophil-like than HL60” couldn’t see this from Figure1.
Line 319-321: how are the numbers calculated for all clones?
Line 327-328: what does CEBPE and PU.1 mean?
Line 319-347: it would be better to have a Venn diagram for all the numbers of differentially expressed genes.
Line 401: use mock samples
Figure 3: would be better to show some of the house-keeping genes.
Round 2
Reviewer 3 Report
The authors have done a good job responding to every point addressed. They defended their work well and the additions and changes they made in the manuscript rendered the motivation and design better explained, and the results more clearly represented.
The manuscript has been greatly improved, and seems good for publication in Genes.